

# DISNET: a framework for extracting phenotypic disease information from public sources

Gerardo Lagunes-García[1], Alejandro Rodríguez-González[1,2], Lucía Prieto-Santamaría[1], Eduardo P. García del Valle[1], Massimiliano Zanin[1] and Ernestina Menasalvas-Ruiz[1]

[1] Centro de Tecnología Biomédica, Universidad Politécnica de Madrid, Pozuelo de Alarcón, Madrid, Spain
[2] Escuela Técnica Superior de Ingenieros Informáticos, Universidad Politécnica de Madrid, Boadilla del Monte, Madrid, Spain

## ABSTRACT

**Background**. Within the global endeavour of improving population health, one major challenge is the identification and integration of medical knowledge spread through several information sources. The creation of a comprehensive dataset of diseases and their clinical manifestations based on information from public sources is an interesting approach that allows one not only to complement and merge medical knowledge but also to increase it and thereby to interconnect existing data and analyse and relate diseases to each other. In this paper, we present DISNET (http://disnet.ctb.upm.es/), a web-based system designed to periodically extract the knowledge from signs and symptoms retrieved from medical databases, and to enable the creation of customisable disease networks.

**Methods**. We here present the main features of the DISNET system. We describe how information on diseases and their phenotypic manifestations is extracted from Wikipedia and PubMed websites; specifically, texts from these sources are processed through a combination of text mining and natural language processing techniques.

**Results**. We further present the validation of our system on Wikipedia and PubMed texts, obtaining the relevant accuracy. The final output includes the creation of a comprehensive symptoms-disease dataset, shared (free access) through the system's API. We finally describe, with some simple use cases, how a user can interact with it and extract information that could be used for subsequent analyses.

**Discussion**. DISNET allows retrieving knowledge about the signs, symptoms and diagnostic tests associated with a disease. It is not limited to a specific category (all the categories that the selected sources of information offer us) and clinical diagnosis terms. It further allows to track the evolution of those terms through time, being thus an opportunity to analyse and observe the progress of human knowledge on diseases. We further discussed the validation of the system, suggesting that it is good enough to be used to extract diseases and diagnostically-relevant terms. At the same time, the evaluation also revealed that improvements could be introduced to enhance the system's reliability.

Corresponding author
Alejandro Rodríguez-González, alejandro.rg@upm.es

## INTRODUCTION

In 1796, Edward Jenner found an important link between the variola virus, which affected only humans and was highly lethal, and the bovine smallpox virus, which attacked cows and was transmitted to humans by physical contact with infected animals, and which, despite its severity, rarely resulted in death. He found that people who became infected with the latter (also called cowpox) did not subsequently catch the former; and thus, that something in the bovine smallpox virus made humans immune to variola virus. This led him to thoroughly investigate the relationship between these diseases and understand their behaviour for more than twenty years; to be finally able to find a cure for the variola virus, saving thousands of humans lives worldwide.

This discovery illustrates the importance of the knowledge that we can get from diseases and, more specifically, from how they are related. Despite the fact that in the last 200 years our understanding of diseases has greatly increased, and valuable advances have been made in this area (*Botstein & Risch, 2003*), the number of those without treatment or cure is still extremely high (e.g., Alzheimer's disease, small cell lung cancer, HIV, etc.). It is thus imperative to explore new approaches and tools to tackle them and, therefore, improve the health of the world's population.

It is almost a truism that the search for new drugs requires a better understanding about diseases. This includes finding new insights on the relationship between diseases (which diseases are related and how), as well as the creation of public and easy-to-access large databases of diseases knowledge (*Pérez-Rodríguez et al., 2019*). During the last decade, such endeavour has been vastly facilitated by the World Wide Web. On one hand, it is possible to find free biomedical vocabularies like Unified Medical Language System (UMLS) (*Bodenreider, 2004*), Human Phenotype Ontology (HPO) (*Robinson et al., 2008*; *Köhler et al., 2017*), Disease Ontology (DO) (*Schriml et al., 2012*) or MeSH (*Lipscomb, 2000*), all of them offering disease classifications, disease coding standards and associated medical resources. On the other hand, one can find bioinformatic databases created by complex medical systems, like DiseaseCard (*Oliveira et al., 2004*; *Dias et al., 2005*; *Lopes & Oliveira, 2013*), MalaCards (*Rappaport et al., 2013*; *Rappaport et al., 2014*; *Espe, 2018*), GeneCard (*Safran et al., 2002*), Diseases Database (DD) (H *Duncan, 2019*, p. 2), DISEASES (*Pletscher-Frankild et al., 2015*), SIGnaling Network Open Resource (SIGNOR) (*Perfetto et al., 2016*), Kyoto Encyclopedia of Genes and Genomes (KEGG) (*Kanehisa & Goto, 2000*), MENTHA (*Calderone, Castagnoli & Cesareni, 2013*), PhosphositePlus (*Hornbeck et al., 2015*), PhosphoELM (*Hornbeck et al., 2015*), UniProtKB (*UniProt Consortium, 2014*), Human Gene Mutation Database (HGMD) (*Stenson et al., 2014*), Comparative Toxicogenomics Database (CTD) (*Mattingly et al., 2006*), and the database for Pediatric Disease Annotation and Medicine (PedAM) (*Jia et al., 2018*). These datasets have generally been created by processing the information from several sources, and they usually offer simple search engines; yet, not all of them have a systematic and structured form of sharing their knowledge. In this context, it is important to relate the quantity of available medical sources and systems on one hand, and the need of health professionals for quality information on the other, helping them performing their work with higher precision and
lower time (*Russell-Rose, Chamberlain & Azzopardi, 2018*). Therefore, diagnostic systems (*Chen et al., 2018*) have become more relevant and researchers such as *Xia et al. (2018)* attempt to take on the challenge through the mining of information from sources such as DO, Symptom Ontology (SYMP) and MEDLINE/PubMed citation records. We can also observe in the literature a large volume of studies that use the mining of texts from different unstructured or semi-structured medical information sources (*Frunza, Inkpen & Tran, 2011*; *Mazumder et al., 2016*; *Singhal, Simmons & Lu, 2016*; *Xu et al., 2016*; *Tsumoto et al., 2017*; *Sudeshna, Bhanumathi & Hamlin, 2017*; *Aich et al., 2017*; *Gupta et al., 2018*; *Rao & Rao, 2018*; *Zhao et al., 2018*); (*Bou Rjeily et al., 2019*).

There is no doubt that the large amount of available bioinformatic resources allows one to both enhance the research in the biomedical field and to have a better understanding of how the diseases behave and how can we fight them. However, most of the already mentioned sources are mainly focused on retrieving and exposing the captured knowledge and are not primarily focused on the analysis of the interactions and relationships that exists between different diseases or different disease characteristics.

In this context, several works have attempted to understand these relationships by creating and analysing disease networks. The complexity of such endeavour was soon clear, as diseases may share not only symptoms and signs, but also genes, proteins, causes and, in many cases, cures (*Goh et al., 2007*; *Zanzoni, Soler-López & Aloy, 2009*; *Barabási, Gulbahce & Loscalzo, 2011*; *Lee et al., 2011*; *Zhou et al., 2014*; *Chen et al., 2015*; *Quwaider & Alfaqeeh, 2016*; *Piñero et al., 2017*; *Lo Surdo et al., 2018*; *Hwang et al., 2019*; *Szklarczyk et al., 2019*; *García del Valle et al., 2019*). One of the most important works on the subject was published in 2007 by *Goh et al. (2007)*, in which the HDN (Human Disease Network) was developed, a network of human diseases and disorders that links diseases based on their genetic origins and biological interactions. Different diseases were then associated according to shared genes, proteins or protein interactions. The hypothesis that different diseases, with potentially different causes, may share characteristics allows the design of common strategies regarding how to deal with the diagnosis, treatment and prognosis of a disease.

Within this line of research it is worth mentioning the Human Symptoms-Disease Network (HSDN) (*Zhou et al., 2014*), an HDN network in which similarities between diseases were estimated through common symptoms. This is an important change in perspective with respect to previous works, in which the focus was centred on the genetic and biological origin of the diseases. In *Zhou et al. (2014)*, diseases are defined by their clinical phenotypic manifestations, i.e., signs and symptoms; this is not surprising, as these manifestations are basic medical elements, and crucial characteristics in the diagnosis, categorization and clinical treatment of the diseases. It was then proposed to use these as a starting point to understand the existing relationships between different diseases.

Building on top of these previous works and stemming from the necessity of having exhaustive and accurate sources of disease-based information, in this paper we present the DISNET (Diseases Networks) system. DISNET aims at going one step further in improving human knowledge about diseases, not only by seeking and analysing the relations between

them, but most importantly, by finding real connections between diseases and drugs, thus potentially enabling novel drug repositioning strategies.

Therefore, the objectives of this research work are:

- Present the first version of the web-based DISNET (phenotypic information) system.
- Describe the characteristics of its retrieval and generation process of phenotypic knowledge.
- Provide an indicator of the accuracy of the information generated by DISNET, through a manual information validation process.
- Provide free access to the DISNET dataset with structured information about diseases and symptoms through the system's API.

The current version of the DISNET system is focused on phenotypic information and allows to capture knowledge about diseases from heterogeneous textual sources. We have five main advantages with respect to the previously described research. Firstly, the use of Wikipedia as the main source of knowledge. While this encyclopaedia has been the object of study of numerous research works, to the best of our knowledge DISNET is the first system to mine texts that describe how the disease manifests itself, and to recover disease codes, leading to a more extensive mapping between several biomedical information sources. Secondly, DISNET offers a public API, that enables the free and structured sharing of the knowledge generated by the system; it is worth noting that having an appropriate method for information sharing, while being a basic element, is not common among the previously reviewed systems. Thirdly, the proposed system allows to recover the temporal evolution of phenotypic information. This is especially relevant for sources like Wikipedia, which is constantly edited, and whose medical articles are frequently corrected and updated. This allows an analysis of the dynamics of diseases, in terms of how their description has been evolving through a collective effort. Fourthly, DISNET has been designed to be able to store and integrate information from heterogeneous sources; this allows to cross-validate and enrich medical knowledge of diseases and symptoms. Future content to be introduced includes genetic and drug information to create a complex multilayer network, where each layer represents the different type of information (phenotypical, biological, drugs). Finally, we also provide an evaluation of the DISNET extracted content, with examples on how diseases can be analysed and their relationships described through a network structure.

In synthesis, it is important to note that for the DISNET platform we have been inspired by the usefulness of some features of relevant systems found during a literature review (*García del Valle et al., 2019*). In some cases, these features have been merged or addressed using a different approach. Therefore, the elements that make DISNET unique are: the capability to include textual biomedical knowledge from several information sources with different structures; the ability to automatically mine each of the included sources, in order to maintain a constant flow of data injection over time, allowing the creation of knowledge captures at different time points; although the system currently has only one NLP tool, the system has the ability to increase the amount of NLP processes used; the free availability of the data generated by the platform, allowing the DISNET results to be exploited by others; and, finally, the implementation of techniques such as RDF to provide another

mechanism for sharing information; DISNET also store all the information related to how the knowledge has been generated, in other words, thanks to this, it is possible to perform a tracking of the generation of knowledge and be able to see where, how and when the data came from, and even to repeat better NLP processes over the same data; and finally, through the set of data obtained by DISNET available free of charge, opens up a range of possibilities, as it allows the creation of disease networks, the application of different analysis techniques or their use in other biomedical or bioinformatics systems.

Beyond this introduction, this paper is organised as follows: 'Materials & Methods' explains the technologies used in the creation of DISNET phenotypical features repository. 'Results' presents the main results obtained in the validation of the system and discussion about them, describes several simple use cases. Finally, 'Discussion' draws some conclusions and discusses future work.

## MATERIALS & METHODS

This section discusses the technical characteristics of the DISNET system, focusing on two aspects: the sources of information hitherto considered, and the DISNET workflow. More specifically, the last point describes how the system retrieves phenotypic information, in the form of raw texts, from the discussed sources; how these texts are processed to obtain diagnostic terms; and how these terms are validated to compile a final list of valid symptom-type terms. The study was approved by the Ethics Committee of the Universidad Politécnica de Madrid. The source code of the entire DISNET platform and their components is available online (*DISNET, 2019k*).

### Information source

As it has previously been shown, it is customary for works aimed at unveiling relationships between diseases to focus on single source of information, in most cases just *abstracts* of Medline articles. On the other hand, the proposed system aims at obtaining inputs from as many sources as possible, to guarantee the recovery of as much knowledge as possible. By bringing together information from different sources, we expect them to complement each other, creating a network with a higher capacity of relating diseases. The rationale for this is that the different sources of textual knowledge, such as Wikipedia or PubMed, are written in different styles and by people with different backgrounds; the information they contain may therefore be complementary. In order to take advantage of such richness, the DISNET system allows the user to query the symptoms according to different rules: for instance, from one or multiple sources, by applying filters based on prevalence information, or on percentages of similarity among others. This clearly comes at a cost: the system should be flexible enough to be able to process sources with different structures. In the remainder of this Section we discuss the patterns used to select data sources, how they have been mined, and finally the challenges involved in such tasks.

### Source selection

Traditionally, in order to obtain the whole body of knowledge that mankind has accumulated about a given disease, one would refer to medical books. Although books

usually contain much of the information available, they also present some important limitations: they are not constantly updated; the automatic access to their content is difficult, especially when digital versions are not available; and they are usually written for study, thus the information they contain is not structured for data mining tasks. On the other hand, one has the World Wide Web, whose main characteristic is to be (mostly) free accessible to anyone with an internet connection. It mainly offers three sources of information. Firstly, the abstract, and in some cases, the full text, of medical papers, which can be accessed through platforms like PubMed. Secondly, specialized sources of information, such as MedlinePlus, MayoClinic, or CDC. Finally, good medical data can be obtained in sources of knowledge that are not specialized, such as Wikipedia or Freebase. Note that all of them have different characteristics, in terms of comprehensiveness, degree of structure of the information, and up-to-datedness (*García del Valle et al., 2019*).

The criteria used for the selection of the sources of information in DISNET are: (i) open access, (ii) recognised quality and reliability, and (iii) availability of substantial quantities of data (structured or not). This suggested to include the following two sources in the system, which are described below: (i) Wikipedia and (ii) PubMed. It is important to note that the system is not closed; on the contrary, thanks to its flexibility, new sources could (and ill) be incorporated in the future.

## Wikipedia

Wikipedia is an online, open and collaborative source of information. It was created by the Wikimedia Foundation and its English edition is the largest and most active one. The monumental and primary task of editing, revising and improving the quality of all articles is not performed by a core of administrators: it is instead the collaborative result of thousands of users. Consequently, this encyclopaedia is considered the greatest collective project in the history of humanity (*Mehdi et al., 2017*). There are currently many initiatives that aim to ensure that the editions in Wikipedia related articles are of high quality (*Azer, 2014*; *Hodson, 2015*; *Matheson & Matheson, 2017*; *Murray, 2019*). Some, for example, have allowed senior medical practitioners to edit some Wikipedia articles, resulting in more stable and high quality texts (*Moturu & Liu, 2009*; *Head & Eisenberg, 2010*; *Cohen, 2013*; *Hasty et al., 2014*; *Temple & Fraser, 2014*; *Farič & Potts, 2014*; *Azer, 2015*; *Azzam et al., 2017a*; *Shafee et al., 2017*; *Sciascia & Radin, 2017*; *Brigo & Erro, 2018*; *Del Valle et al., 2018*).

Wikipedia contains more than 155,000 articles in the field of medicine (*Azzam et al., 2017b*) and is one of the most widely used medical sources (*Friedlin & McDonald, 2010*) by the general community (*Aibar, 2017*) and also by medical specialists (*Azer, 2014*; *Shafee et al., 2017*), the latter ones having deeply been involved in its enrichment (*Azzam et al., 2017b*; *Cohen, 2013*). One of the initiatives is the Cochrane/Wikipedia, which aims at increasing reliability in articles with medical content (*Matheson & Matheson, 2017*). In 2014 Wikipedia was referred to as "*the single leading source of medical information for patients and health care professionals*" by the Institute of Medical Science (IMS) (*Heilman & West, 2015*). This stems from the fact that an increasing number of people in the medical field are becoming aware of the importance of collaborating and generating quality content in the world's largest online encyclopaedia.
We have focused on Wikipedia in its English edition, and specifically on those articles categorized as diseases. In order to obtain a list of such articles we resort to conventional DBpedia and DBpedia-Live (DBpedia), an open and free Web repository that stores structured information from Wikipedia and other Wikimedia projects. By containing structured information, this source allows complex questions to be asked through SPARQL queries (*SPARQL Query Language for RDF, 2017*). We developed a query (*DISNET, 2018a*) that is able to get all the articles of Wikipedia in English referring to human diseases and run it in the **Virtuoso environment SPARQL Query Editor of DBpedia** (*OpenLink Software, 2019*). This first approach to detecting and extracting Wikipedia's web links can be addressed in different ways and in the Discussion and conclusions section we will talk about them.

Even though disease articles have a standard structure, due to the very nature of Wikipedia, articles can be edited by anyone; consequently, it is possible to find articles that do not comply with the standard form that the creators of the encyclopaedia propose (*Wikipedia, 2018*). The structure is organized in sections, of which we have selected those whose content is related to the phenotypic manifestations of the disease. The essential sections mined by DISNET are: "*Signs and symptoms*", "*signs and symptoms*", "*Symptoms and causes*", "*Signs*", "*Symptoms*", "*Causes*", "*Cause*", "*Diagnosis*", "*Diagnostic*", "*Causes of injury*", "*Diagnostic approach*", "*Presentation*", "*Symptoms of …* ", "*Causes of …*" , and *infobox.*

The data retrieved from these sections are: (i) the texts (paragraphs, lists and tables) contained in the previously described sections; (ii) the links contained in these texts; and (iii) the disease codes of vocabularies external to Wikipedia, which can be found in the *infoboxes* of the article. Note there are two types of *infobox*. Figure 1 shows an example of the external vocabulary codes retrieved in a vertical *infobox*, usually located at the beginning of the document; Fig. 2 shows an example of a horizontal *infobox*, generally located at the foot of the document. These disease codes in different vocabulary are relevant elements when searching for diseases in the system's database. The list of external vocabularies to DISNET can be found at (*DISNET, 2018b*).

## PubMed

PubMed comprises more than 28 million biomedical literature citations from MEDLINE, life science journals and online books. Quotations may include links to full text content from PubMed Central (*NCBI, 2019*) and editorial websites (*pubmeddev, 2019*). As in other studies, we here only considered the abstracts of the articles, as, firstly, it is not always possible to access the full text, and secondly, the full text of articles does not follow a standard format. However, we are aware of the limitations of the extraction of information only for abstracts (*Westergaard et al., 2018*), and future versions of DISNET platform will focus in extracting the content from the full paper when possible. Note that in PubMed the information about one single disease is spread among multiple documents—as opposed to Wikipedia, in which there is a bijective relationship between articles and diseases.

Obtaining the list of diseases in PubMed involves two main steps. Firstly, one should extract the list of MeSH terms (DMTL) relating to human diseases $C$, which are categorized
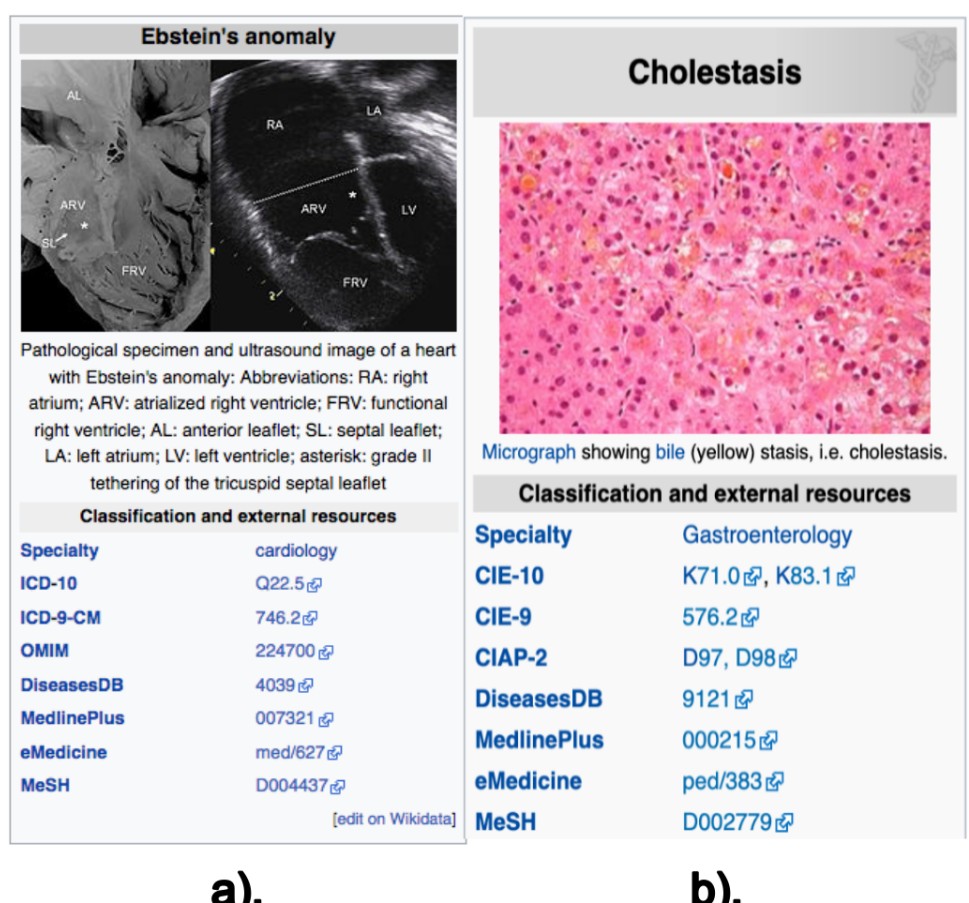

**Figure 1** Screenshot of the Wikipedia content of infoboxes in two diseases. Two instances of infoboxes, i.e., the top right part of a Wikipedia page mainly containing the identifiers of the disease in external vocabularies; these screenshots are for the articles on (A) Ebstein's anomaly and (B) Cholestasis. Images licensed under CC BY SA 3.0.

from *C01* to *C20* (excluding those categories such as "Animal Diseases" or "Wounds and Injuries") as shown in the classification tree in Fig. 3 (*United States National Library of Medicine, 2019*); and map each disease with Human Disease Ontology (*OBO Foundry, 2019*) to obtain disease codes of the vocabulary ICD-10, OMIM, MeSH, SNOMED_CT and UMLS. Note that the use of multiple vocabularies aims at obtaining the greatest amount of means (identified codes) to identify diseases in different sources of information. As a second step, it is necessary to extract all PubMed articles whose terms are associated with each of the elements of the previously extracted disease list DMTL, through PubMed's Entrez API (AEPM) it is possible to carry out this task, because this allows access to all Entrez databases including PubMed, PMC, Gene, Nuccore and Protein. An important feature to mention of the AEPM, and also used in our work, has been the sorting of articles by their relevance (*Fiorini et al., 2018*), managing to focus the efforts on those articles with better quality. Thus, this configuration has given us the possibility to obtain, if they exist, the 100 most relevant articles of each MeSH term consulted. Specifically, for each article
### a). Influenza disease codes

| Classification | **ICD-10**: J10 ⧉, J11 ⧉ · **ICD-9-CM**: 487 ⧉ · **OMIM**: 614680 ⧉ · **MeSH**: D007251 ⧉ · **DiseasesDB**: 6791 ⧉ | D |
| --- | --- | --- |
| **External resources** | **MedlinePlus**: 000080 ⧉ · **eMedicine**: med/1170 ⧉ ped/3006 ⧉ · **Patient UK**: Influenza ⧉ | |

### b). Cancer disease codes

| Classification | **ICD-10**: C00-C97 ⧉ · **ICD-9-CM**: 140 ⧉ —239 ⧉ · **MeSH**: D009369 ⧉ · **DiseasesDB**: 28843 ⧉ | D |
| --- | --- | --- |
| **External resources** | **MedlinePlus**: 001289 ⧉ | |

**Figure 2 Information located at the bottom of Wikipedia articles.** The codes are divided into two sections: (i) classification codes of vocabulary type sources and (ii) codes in external data sources. The two screenshots are for the articles on (A) influenza and (B) cancer. Image licensed under CC BY SA 3.0.

we retrieve: (1) abstract, (2) authors' names, (3) unique identifier in PubMed and PubMed Central, (4) doi (digital object identifier), (5) title, (6) associated MeSH terms and (7) keywords. The workflow for extracting texts from PubMed documents is shown in Fig. 4.

## Challenges

Mining information from the sources previously described entails several computational challenges, which may be boiled down to one requirement for the DISNET system: the need of a high versatility in data acquisition. We here review such challenges, as these partly explain the adopted software solution.

First of all, the mapping disease-webpage may take different forms. Specifically, it is one to one for Wikipedia, as all the information of a disease is included in a single page; but it becomes one to many for PubMed, in which multiple articles are available for each single concept. Consulting the latter thus requires a more complex procedure.

Secondly, and as one may expect, the specific structure of each source of information is different—i.e., a page of Wikipedia has not the same structure of a PubMed article. This requires further flexibility, in terms of the development of a modular structure with specific crawlers for each source.

Finally, it is worth noting that, while here we have only considered texts, much information is available in different medias, like images, videos and others binary files. While not implemented at this stage, the system should be flexible enough to accommodate such sources in the future.

## Data retrieval and knowledge extraction

This section describes the general architecture of the DISNET system, including the data extraction and the subsequent knowledge extraction. In the sake of clarity, such architecture is further depicted in Fig. 5.

Diseases [C] ⊖
    Bacterial Infections and Mycoses [C01] ⊕
    Virus Diseases [C02] ⊕
    Parasitic Diseases [C03] ⊕
    Neoplasms [C04] ⊕
    Musculoskeletal Diseases [C05] ⊕
    Digestive System Diseases [C06] ⊕
    Stomatognathic Diseases [C07] ⊕
    Respiratory Tract Diseases [C08] ⊕
    Otorhinolaryngologic Diseases [C09] ⊕
    Nervous System Diseases [C10] ⊕
    Eye Diseases [C11] ⊕
    Male Urogenital Diseases [C12] ⊕
    Female Urogenital Diseases and Pregnancy Complications [C13] ⊕
    Cardiovascular Diseases [C14] ⊕
    Hemic and Lymphatic Diseases [C15] ⊕
    Congenital, Hereditary, and Neonatal Diseases and Abnormalities [C16] ⊕
    Skin and Connective Tissue Diseases [C17] ⊕
    Nutritional and Metabolic Diseases [C18] ⊕
    Endocrine System Diseases [C19] ⊕
    Immune System Diseases [C20] ⊕
    Disorders of Environmental Origin [C21] ⊕
    Animal Diseases [C22] ⊕
    Pathological Conditions, Signs and Symptoms [C23] ⊕
    Occupational Diseases [C24] ⊕
    Chemically-Induced Disorders [C25] ⊕
    Wounds and Injuries [C26] ⊕

**Figure 3**   Classification tree of diseases according to MeSH.

## The extraction process

The first step of the DISNET pipeline is in charge of retrieving the information from the sources previously identified and described. For each one of this, and before running the actual web crawler, the ''Get Disease List Procedure'' (GDLP) component is responsible for obtaining the list of diseases to be mined, thus providing links to all available disease related documents. For example, the GLDP associated to Wikipedia articles makes use of the SPARQL query (*DISNET, 2018a*); similarly, the links for the PubMed's articles are retrieved through a list of MeSH terms.

Once the URL list has been collected, the ''Web Crawler'' (WC) module is in charge of connecting to each of the hyperlinks and extracting the specific text that describes the phenotypical manifestations, as well as the links (references) contained within the texts (*Hedley, 2019*). In addition, and whenever possible, it attempts to extract information related to the coding of diseases, i.e., the codes used to identify the disease in different databases or existing data vocabularies. Currently it is able to retrieve information from

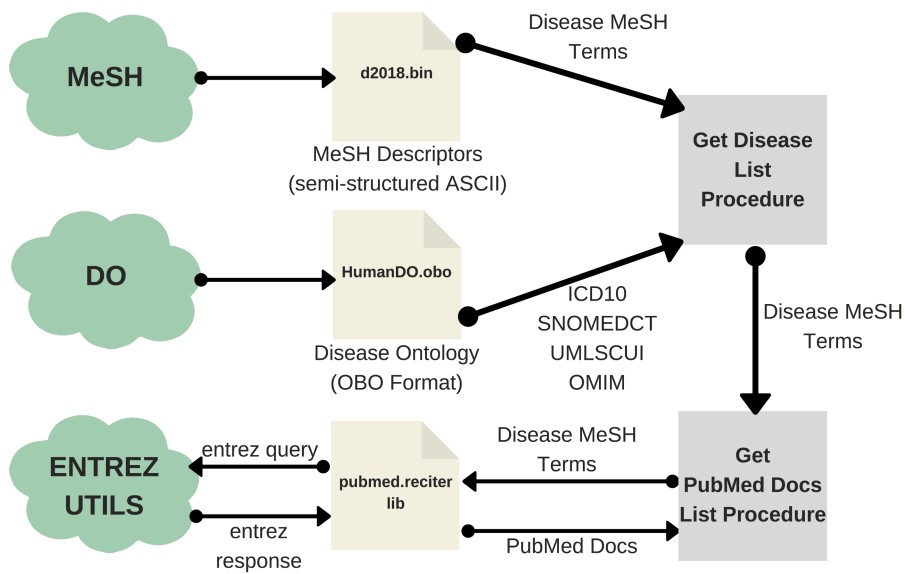

**Figure 4    Workflow of the text extraction procedure for PubMed.**

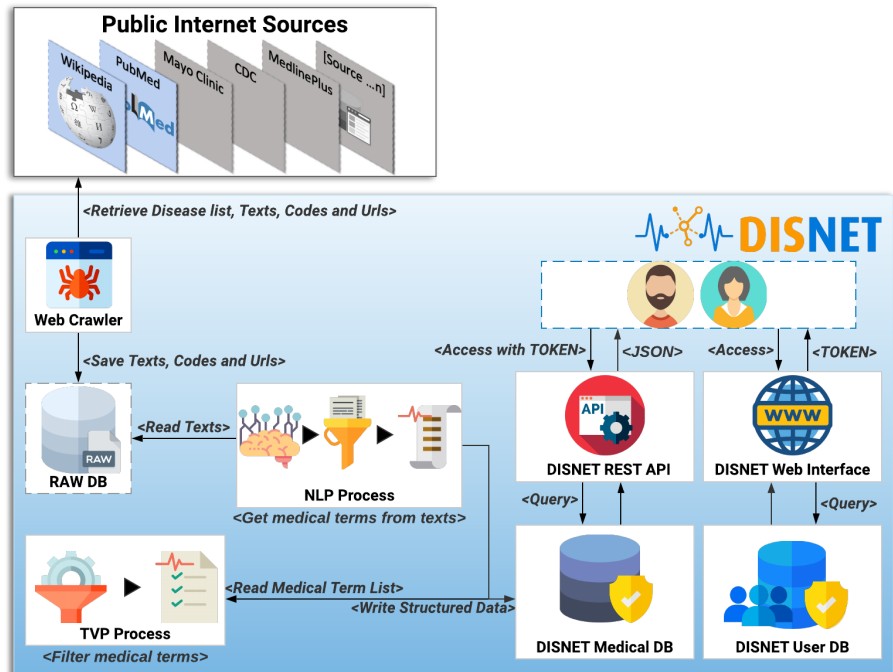

**Figure 5    DISNET Architecture/Workflow.** Image credits: Nohat, licensed under CC BY-SA 3.0 (https://es.wikipedia.org/wiki/Wikipedia#/media/Archivo:Wikipedia-logo-v2.svg); Smashicons, prettycons, Freepik, Dimitry Miroliubov, Becris, Icon Pond, and Prosymbols from Flaticon.

more than 6,692 articles in Wikipedia and from 229,160 article abstracts in PubMed. The information mined by WC is stored in an intermediate database called "Raw DB", which contains the raw unprocessed text.

The next step within the pipeline is called "NLP Process" (NLPP). This component is responsible for: (i) reading all the texts of a snapshot, and (ii) obtaining for each text a list of relevant clinical concepts/terms, discarding any unrelated paragraphs or words. At the moment NLPP uses MetaMap (*Aronson, 2001*; *Rodríguez González et al., 2018*) as a Natural Medical Language Processing tool to extract clinical terms of interest –see online NLP Tools and Configuration section (*DISNET, 2019a*). Semantic types (SM) are important elements created by UMLS to define categories of concepts. MetaMap uses SM to find medical elements, and a full list of them is available online (*United States National Library of Medicine, 2018*).

The output of the NLP process is stored in the "DISNET Medical DB" (DMDB) database. It stores, in a structured way, the medical concepts that have been obtained by the NLPP, as well as any information required to track the origin of such concepts –in order to track any error that may later be detected. Therefore, and to summarize, the information stored in a structured way in DMDB is: (i) the medical concepts with their location, information and semantic types, (ii) the texts from which they were extracted and the links by them contained, (iii) the sections which the texts belong to, (iv) the document or documents describing the disease (Web link) and (v) the disease identifiers codes in different vocabulary or databases. Additional information, as the day of the extraction and the source, is further saved.

Before reaching the last step of the process, it is important to highlight the nature of the information hitherto stored. Specifically, the system has not extracted only signs or symptoms of a disease, but instead medical terms that we believe may be phenotypic manifestations of disease. It is thus necessary to filter those that are not relevant for the objective initially described.

Having clarified this, the next component of the pipeline, the "TVP Process" TVPP, reads all the concepts of a snapshot - source pair and filters them. This process is responsible for determining whether these UMLS medical terms are really phenotypic manifestations, and for storing the results back in the DMDB. TVPP is based on the Validation Terms Extraction Procedure that was developed, implemented and tested by *Rodríguez-González et al. (2015)*. The results of this component (a purification of concepts) are thus those validated terms that we will consider as true phenotypic manifestations of diseases.

The DISNET extraction process (IEPD), i.e., the process of retrieving and storing information about diseases, basically ends here. Nevertheless, for the sake of providing an accessible and user-friendly way of retrieving and manipulating this information, DISNET also offers a REST-based interface. This is described in detail in the system website (http://disnet.ctb.upm.es/apis/disnet); also refer to 'Discussion' for an application example.

## RESULTS

This section describes how the medical concepts data set is built, for then validating and analysing its content.

## Construction of the DB

The database in the DISNET system contains information recovered from three sources of information: Wikipedia and PubMed. From Wikipedia we have 26 snapshots, from February 1st, 2018 to February 15th, 2019, for PubMed we have one snapshot, that of April 3rd, 2018. Within the system it is possible to consult, for each snapshot and source, the total number of articles with medical terms, the total number of medical terms found, the number of processed texts, the total number of retrieved codes, and the total number of semantic types found (*DISNET, 2019b*).

When summing that sources, the system counts with 6,545 diseases, 2,212 medical terms from UMLS (SNOMED-CT) and 19 semantic types, which can be consulted online (*DISNET, 2019c*).

Wikipedia snapshots are built using the configurations that are available online (*DISNET, 2019d*). We have obtained a list of 11,074 articles catalogued as diseases in Wikipedia according to DBpedia (*DISNET, 2019e*), from which we obtained 6,692 articles with at least one text referring to phenotypic knowledge of the disease, or at least one code to an external information source, 4,798 of which were found to be relevant medical concepts (*DISNET, 2019f*).

The snapshot for PubMed has been built using the configuration described online (*DISNET, 2018c*). This snapshot has been built on top of a list of 2,354 MeSH terms (*DISNET, 2018c*) referring to human diseases, but only for 2,213 MeSH terms did we obtain information (199,013 scientific articles in total, i.e., about 0.71% of the 28 million articles existing in PubMed (*DISNET, 2018d*)) and of each of these PubMed articles obtained, only in 174,900 were abstracts found and only in 125,515 were relevant medical terms found. Figures 6 and 7 presents some basic database statistics at an aggregated level as well as by source (for Wikipedia and PubMed). Some notable differences can be observed; for instance, the five most common terms for Wikipedia are *Pain*, *Lesion*, *Neoplasms*, *Magnetic resonance imaging*, *Inflammation* and *Malnutrition*, while for PubMed these are *Neoplasms*, *Lesion*, *Magnetic resonance imaging*, *Malnutrition* and *Inflammation*. Similarly, the three diseases with the greatest number of concepts in Wikipedia are *Kawasaki disease*, *Cerebral palsy* and *Hypoglycemia,* while for PubMed these are *Hypercalcemia*, *Cranial nerve palsy* and *Beriberi*.

## Data evaluation of the DB

In this section, we discuss the results of the validation process we executed on the system, to ensure the relevance of the diagnostic knowledge (valid medical diagnostic terms) generated through our NLP process (MetaMap and TVP). The evaluation has been made on both Wikipedia and PubMed mined. Our evaluation process has been performed by three people with experience in clinical information, in order to avoid "ties" of identification (or discarding) of elements of diagnostic knowledge (DKEs) during the NLP process.

The validation for Wikipedia was carried out on the February 1st, 2018 snapshot, selecting 100 diseases at random with the only condition of having at least 20 valid medical terms in order to ensure that our validation procedure analyses articles with a high concentration of medical knowledge. Similarly, the validation for PubMed has been done on the April 3rd,

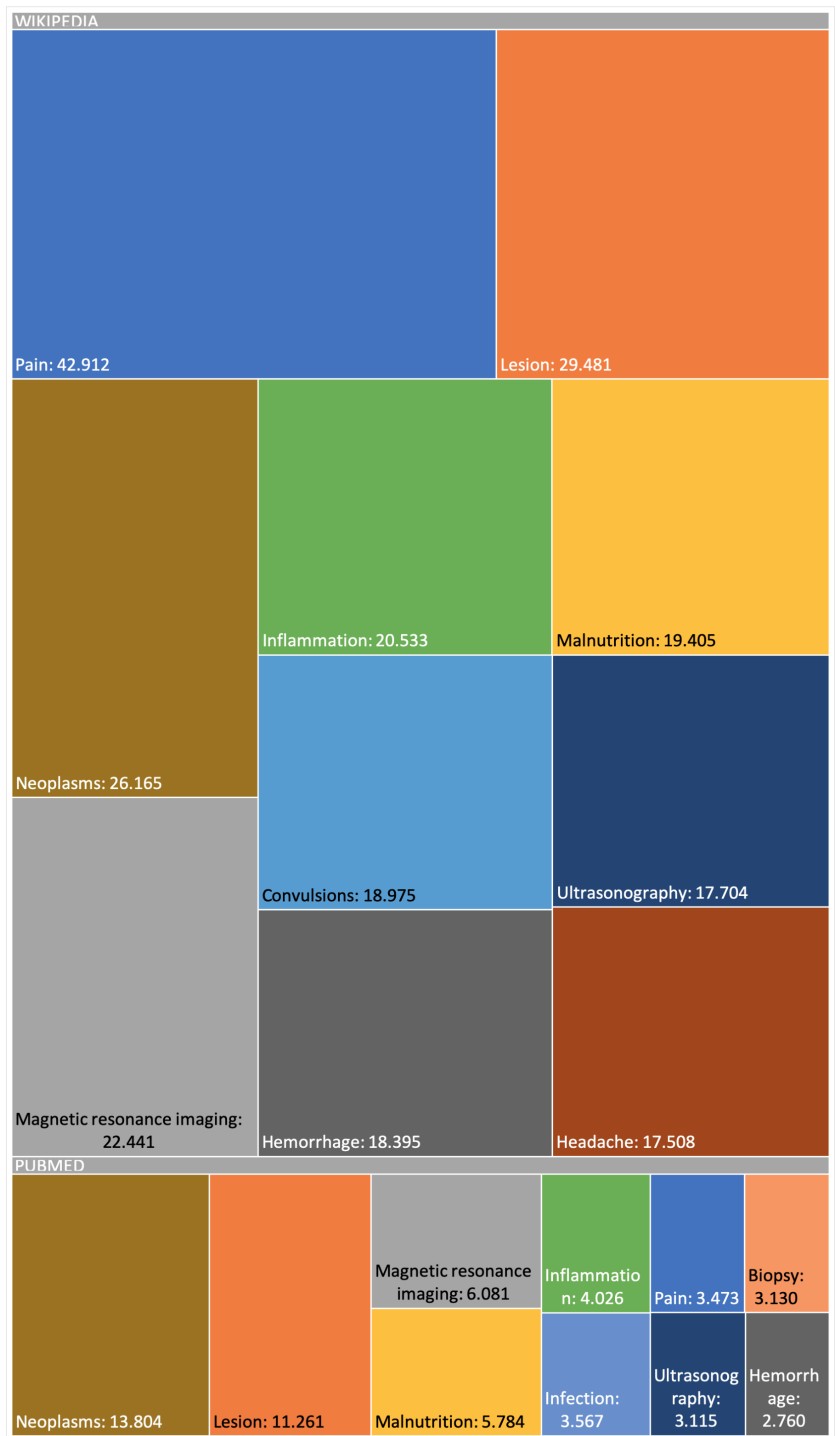

**Figure 6** Appearance frequency of the most common medical terms in both Wikipedia and PubMed.

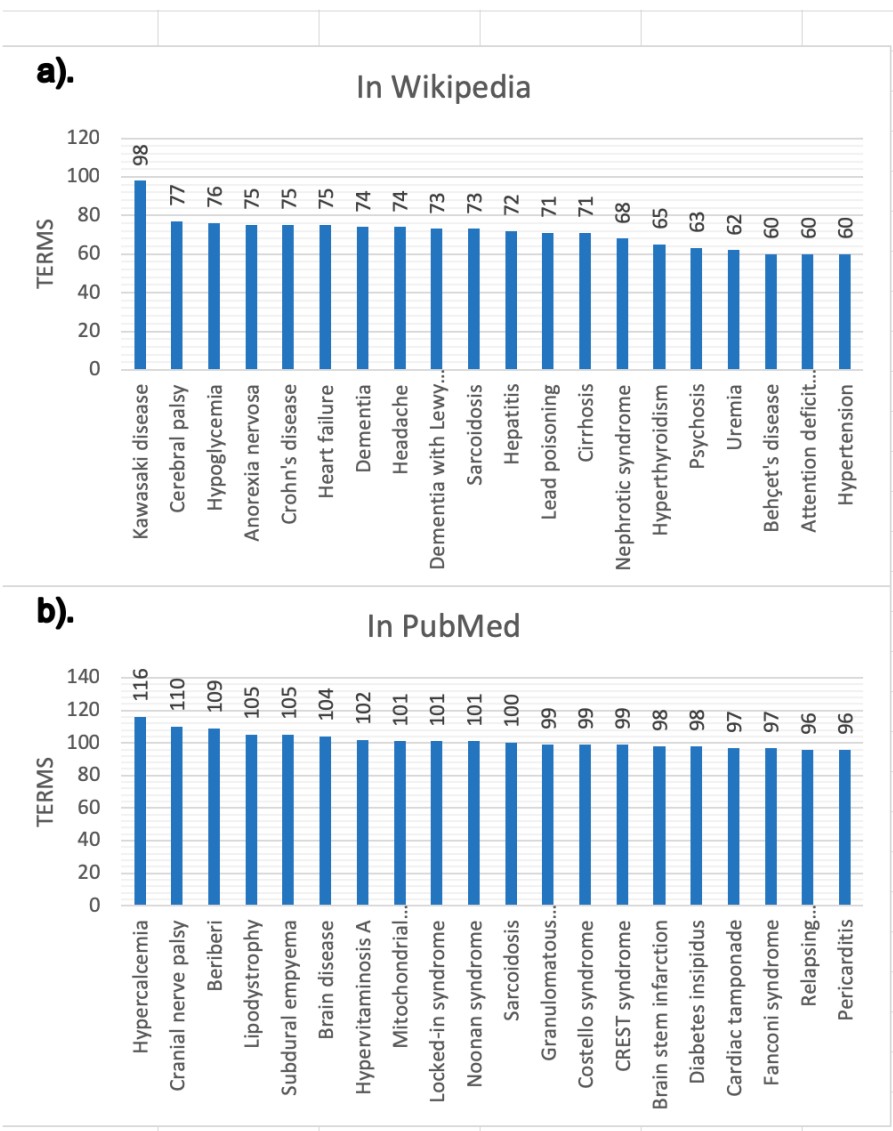

**Figure 7** **Diseases with more validated medical terms.** (A) Results in Wikipedia; (B) results in PubMed.

2018 snapshot, selecting a random sample of 100 article abstracts. It is necessary to highlight that the validation procedure was designed to carry out on articles and due to the nature of each of the sources it is necessary to remember that Wikipedia articles are composed by one or more texts, while PubMed articles are composed by only one text, the abstract. And for this reason for Wikipedia, to validate an article means to validate a disease, for PubMed to validate an article means to validate a part of a disease. These snapshots were performed at different times, and therefore with different configurations –the latter ones can be viewed online (*DISNET, 2018c*). During the validation of Wikipedia, we detected that the initial configuration of MetaMap did not find all the necessary medical concepts: for instance, Anxiety, Stress, Amnesia, Bulimia and other psychological concepts were missing. We
therefore decided to update the initial list of semantic types to be detected (see online NLP Tools and Configuration section (*DISNET, 2019d*)) by adding the following elements: **Intellectual Product**, **Mental Process**, **Mental or Behavioral Dysfunction**, **Pathologic Function**, **Congenital Abnormality**.

The evaluation was conducted through a thorough manual analysis of the basic data. For each disease obtained from Wikipedia or PubMed we compared: (1) the list of medical terms extracted manually from the texts describing the disease; (2) the list of medical terms extracted by MetaMap from the same texts; (3) the value (TRUE=valid or FALSE=invalid) resulting from the TVP process for each term found by MetaMap; (4) the value of diagnostic relevance for a disease for each term. An example of the format of the Acute decompensated heart failure validation sheet for Wikipedia is shown in Fig. 8.

It is possible to note that an additional column was also present, called RELEVANT, and which synthesises all the information available about the relevance of a term to a disease. The possible values of this column are defined as:

1. RELEVANT = **YES**. If (WIKIPEDIA = YES) & (METAMAP = YES) & (TVP = (YES or NO)), that is, it is considered to be a valid medical concept for the diagnosis of a disease.

2. RELEVANT = **NO**. If (WIKIPEDIA = YES) & (METAMAP = YES) & (TVP = NO), that is, it is considered to be a medical concept that is nonspecific, and thus too general to be helpful in the diagnosis of a disease.

3. RELEVANT = **FPREAL**. If (WIKIPEDIA = YES) & (METAMAP = YES) & (TVP = YES). The term **is not relevant** because it is considered to be a nonspecific, general concept that does not make sense for diagnosis, even though MetaMap has detected it and the TVP process has evaluated it as a diagnostic term. For example, in an excerpt from Acute decompensated heart disease on Wikipedia: "*Other cardiac symptoms of heart failure include chest pain/pressure and palpitations…*", MetaMap has detected **Chest pain** and **Pain** from "*chest pain*", both were marked as TRUE by TVP but the concept dismissed by nonspecific and general was Pain.

4. RELEVANT = **FPCONTEXT**. If (WIKIPEDIA = YES) & (METAMAP = YES) & (TVP = YES). The term **is not relevant** because it is outside the diagnostic context, even though MetaMap has detected it and the TVP process has evaluated it as a diagnostic term. In other words, this term has been obtained from texts whose content is outside the diagnostic context. For example, in an excerpt from *Acute decompensated heart failure* disease on Wikipedia: "*Other well recognized precipitating factors include anemia and hyperthyroidism…*", MetaMap has detect **Anemia** and **Hyperthyroidism** which are medical terms but in context we dismiss them because they are risk factors for that disease.

5. RELEVANT = **FN**. If (WIKIPEDIA = YES) & (METAMAP = NO) & (TVP = NO). These terms were manually detected in the texts, but MetaMap failed in recognising them.

The cases (3) and (4) above define situations in which the detected term is esteemed to be of no relevance, and as such represent cases of false positives. It is nevertheless necessary to discriminate the reason behind such error, which can be because: (i) the term is a very

## Acute decompensated heart failure

| WIKIPEDIA TERMS | METAMAP TERMS | DISNET VALIDATION | | | |
|---|---|---|---|---|---|
| NAME | NAME | WIKIPEDIA | METAMAP | TVP | RELEVANT |
| 1 acute, myocardial, infarction | Acute myocardial infarction | YES | YES | YES | FPCONTEXT |
| 2 illness | Illness (finding) | YES | YES | YES | FPREAL |
| 3 hyperthyroidism | Hyperthyroidism | YES | YES | YES | FPCONTEXT |
| 4 anemia | Anemia | YES | YES | YES | FPCONTEXT |
| 5 weightloss | Weight decreased | YES | YES | YES | YES |
| 6 palpitations | Palpitations | YES | YES | YES | YES |
| 7 nausea | Nausea | YES | YES | YES | YES |
| 8 chest, pain | Chest pain NOS | YES | YES | YES | YES |
| 9 exertional, dyspnoea | Dyspnea on exertion | YES | YES | YES | YES |
| 10 pneumonia | Pneumonia | YES | YES | YES | FPCONTEXT |
| 11 high, blood, pressure | Hypertensive disease | YES | YES | YES | FPCONTEXT |
| 12 weakness | Weakness | YES | YES | YES | YES |
| 13 pain | Pain | YES | YES | YES | FPREAL |
| 14 heart, failure | Heart failure | YES | YES | YES | FPCONTEXT |
| 15 paroxysmal, nocturnal, dyspnoea | Paroxysmal nocturnal dyspnea | YES | YES | YES | YES |
| 16 orthopnoea | Orthopnea | YES | YES | YES | YES |
| 17 difficulty, breathing | Dyspnea | YES | YES | YES | YES |
| 18 heart, attack | Myocardial infarction, NOS | YES | YES | YES | FPCONTEXT |
| 19 abnormal, heart, rhythms | Cardiac arrhythmia | YES | YES | YES | FPCONTEXT |
| 20 bloating | Abdominal bloating | YES | YES | YES | YES |
| 21 chest, pressure | Pressure in chest | YES | YES | YES | YES |
| 22 low, urine, output | Oliguria | YES | YES | YES | YES |
| 23 fatigue | Fatigue | YES | YES | YES | YES |
| 24 jugular, venous, distension | Jugular venous engorgement | YES | YES | YES | YES |
| 25 atrial, fibrillation | Electrocardiographic atrial fibrillatio | YES | YES | NO | NO |
| 26 left, ventricular, failure | Left-sided heart failure | YES | YES | NO | NO |
| 27 sign, signs | Physical finding | YES | YES | NO | NO |
| 28 excess, fluid | Fluid overload | YES | YES | NO | NO |
| 29 chronic, heart, failure | Chronic heart failure | YES | YES | NO | NO |
| 30 pressure | Pressure (finding) | YES | YES | NO | NO |
| 31 acute, heart, failure | Acute heart failure | YES | YES | NO | NO |
| 32 myocardial, infarction | Electrocardiogram: myocardial infarction (finding) | YES | YES | NO | NO |
| 33 decompensation | Decompensation | YES | YES | NO | NO |
| 34 gasping | Gasping for breath | YES | YES | NO | NO |
| 35 symptom, symptoms | Symptom | YES | YES | NO | NO |
| 36 confusion | Confusion | YES | YES | YES | YES |
| 37 fluid, retention | Body fluid retention | YES | YES | YES | FPCONTEXT |
| 38 memory, impairment | Memory impairment | YES | YES | YES | YES |
| 39 sensitive | Hypersensitivity | YES | YES | NO | NO |
| 40 anxiety | Anxiety | YES | YES | YES | YES |
| Acute pulmonary edem | | YES | NO | NO | FN |
| loss of appetite | | YES | NO | NO | FN |
| waking up at night to urinate | | YES | NO | NO | FN |
| cerebral symptoms | | YES | NO | NO | FN |

**Figure 8 Sheet validation.** The results for the disease Acute decompensated heart failure according to the Wikipedia snapshot of February 1st, 2018 are shown.

general, nonspecific concept whose definition does not represent and contributes nothing to the diagnosis (FP_REAL), or ii) because the term is a medical term that is out of place
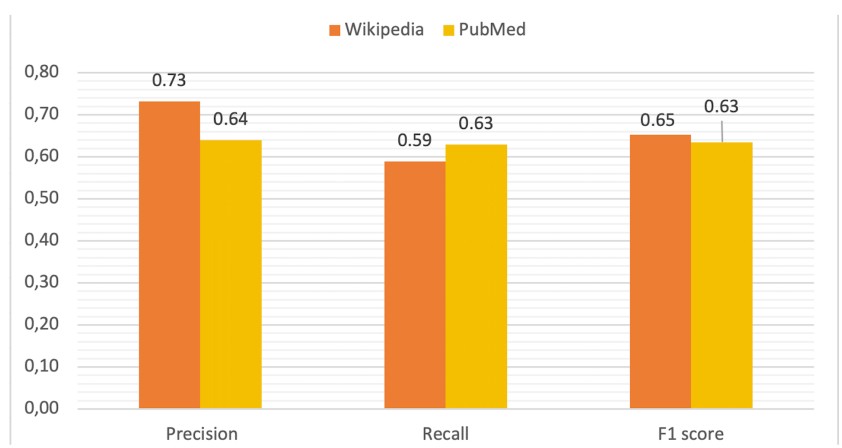

**Figure 9** Comparative of validation metrics in both Wikipedia and PubMed.

with respect to the context that is narrated in the text—in other words, it could be a valid diagnostic term but not for the disease that is under validation or in the context in which have been described and therefore should be discarded (FP_CONTEXT).

Using this information for all diseases and terms, true positive (**TP**), false positive (**FP**), true negative (**TN**) and false negative (**FN**) rates were computed in order to calculate precision, recall and F1 score values as metrics to measure the performance of DISNET system. The mean values for these parameters are depicted in Fig. 9. The **TP** is all terms with (WIKIPEDIA = YES) & (METAMAP = YES) & (TVP = YES) & (RELEVANT = YES). As previously explained, the **FP** is composed of two parts, being the total FP the sum of **FP_REAL** + **FP_CONTEXT**:

- **FP_REAL** = (WIKIPEDIA = YES) & (METAMAP = YES) & (TVP = YES) & (RELEVANT = FPREAL).
- **FP_CONTEXT** = (WIKIPEDIA = YES) & (METAMAP = YES) & (TVP = YES) & (RELEVANT = FPCONTEXT).

**FN** is also composed of two parts, i.e., **FN_METAMAP** + **FN_TVP**.

- **FN_METAMAP** = (WIKIPEDIA = YES) & (METAMAP = NO) & (TVP = NO) & (RELEVANT = FN). These are terms that MetaMap has not found.
- **FN_TVP** = (WIKIPEDIA = YES) & (METAMAP = YES) & (TVP = NO) & (RELEVANT = YES). These are terms that TVP has validated as false while being relevant.

Finally, the **TN** measures the TVP process (WIKIPEDIA = YES) & (METAMAP = YES) & (TVP = NO) & (RELEVANT = NO). In the Table 1 are reported the values obtained for Wikipedia and PubMed.

Detailed results for each disease are available online, for Wikipedia (*DISNET, 2019h*) and for PubMed (*DISNET, 2019g*), including the list of terms manually extracted from the relevant texts of the articles, the matching with the list of terms provided by Metamap, the result of the TVP process for each term and the value of relevance as annotated by our researchers.

**Table 1  Total values from the February 1st, 2018 snapshot of Wikipedia and the April 3rd, 2018 snapshot of PubMed.**

| Parameter | Wikipedia | PubMed |
|-----------|-----------|--------|
| TP | (31.11%) 2,075 | (31.20%) 724 |
| FP | (11.41%) 761 | (17.54%) 407 |
| FPREAL | 279 | 107 |
| FPCONTEXT | 482 | 300 |
| TN | (35.78%) 2,386 | (32.84%) 762 |
| FN | (21.68%) 1,446 | (18.40%) 427 |
| FN_METAMAP | 709 | 201 |
| FN_TVP | 737 | 226 |
| TOTAL | (100%) 6,668 | (100%) 2,320 |
| PRECISION | 0.731 | 0.640 |

Results indicate that our NLP (MetaMap + TVP) process is sufficiently reliable, with an accuracy of 0.731 (confidence interval of [0.710, 0.753], calculated through a Wilson's score interval with continuity correction and a confidence level of 99%) for Wikipedia and of 0.640 (confidence interval of: [0.606, 0.680]) for PubMed (Fig. 9). The results of the calculations of these parameters for each disease can be viewed online for Wikipedia (*DISNET, 2019i*) and for each abstract in PubMed (*DISNET, 2018e*).

About the results for **FP** presented in Table 1, we can say that they are mainly due to the configuration used for MetaMap for the extraction of terms, extended in successive extractions to avoid leaving out terms that are relevant for the detection of diseases.

Thus, one of the last extensions in the search terms added the semantic types Mental or Behavioral Dysfunction and Intellectual Product; thanks to this extension, important symptoms have been detected for certain diseases, which were not detected before, such as: *Anxiety*, *Bulimia*, *Anorexy*, *Stress*, etc. We believe that it is better to discard those terms that are not relevant than to omit those that are relevant to a disease.

It is further interesting to observe the large difference in the false positive rates between Wikipedia (11.41%) and PubMed (17.54%). We speculate that this is due to the concretion of articles. Accordingly, in Wikipedia, articles referring to one disease refer almost exclusively to that particular disease, and thus include no irrelevant terms—with a few exceptions related to differential diagnoses. Nevertheless, this is not the case of PubMed articles as a significant part of them are not so specific. Many are the articles describing real medical cases, where the symptoms are those displayed by a given patient, plus others referring to congenital diseases of the patient, or even diseases that he/she previously possessed. Consequently, the same PubMed article includes symptoms of many different diseases that, although being true medical terms and thus being recognized by MetaMap, are not relevant to the disease under analysis.

For **TN,** we must also take into account that most of the terms extracted by MetaMap as relevant have been purged by TVP, which has been in charge of determining which terms are relevant and which are not, so that the vast majority of terms extracted by MetaMap

that are not relevant to the disease have been classified in this way by TVP (35.78% for Wikipedia and 32.84% for PubMed).

In addition, we have observed that most of the true negative terms in both Wikipedia and PubMed are constant, and include: *indicated*, *syndrome*, *disease*, *illness*, *infected*, *sing*, *symptoms*, *used to*, etc.

Finally, **FN** are those terms that are relevant to the disease in question, but that have not been detected by MetaMap; note that these have been manually extracted for the validation process. The vast majority of **FN** are formed by complex expressions of the language, so their detection is challenging for any NLP tool. We can further observe that the difference in the ratio of false negative between Wikipedia (21.68%) and PubMed (18.40%) is 3.28%. We believe that this difference is mainly due to the forms of expression used in both sources, with Wikipedia being more discursive, as opposed to the scientific style of PubMed.

In synthesis, we can conclude that a clear relationship can be observed between the performance of the system and the nature of the underlying data source. Specifically, while PubMed is an exclusively medical source, created, written and edited by specialists in the field, Wikipedia is a source of public information, written by anyone who has access to the web, so that the articles in it contained can be written by medical students or just users with some knowledge in the field, whose expressions cannot be assimilated to those of specialists who write PubMed. Considering that the tool used by DISNET for the extraction of medical terms (MetaMap) is a medical tool, it is not surprising that it displays a greater capacity for the recognition of medical terms, as opposed to more colloquial terms formed by more complex phrases; thus, there are terms such as ''*Swollen lymph glads under the jaw*'', or ''*sensation of swelling in the area of the larynx*'', that MetaMap cannot recognize.

It is true that the validation percentages do not seem very high, but we must take into account the following facts, firstly, that there is no other system that extracts and generates phenotypic information using an approach as proposed in this document and secondly, the objective of the document is not clinical, but purely research, and thus allows all the knowledge generated to be put within the reach of other researchers and for the scientific community in general. Therefore, the use of DISNET medical information is in the hands of all types of people and they are therefore responsible for the use they give to such data. It is also important to mention that despite the complex and inherent nature of the texts from different sources, the percentages reflect good research work.

## A use case

To illustrate the possible use of the DISNET system, we here present a simple use case, which consists of the creation of several basic DISNET queries, and the visualization of the corresponding results.

The DISNET API has the capacity to create a variety of queries and in this section only a couple of queries have been created in order to provide a small example of the capacity to support research into the proposed system.

## Creation of DISNET queries

For the sake of simplicity, we will here focus on two of the most important characteristics of DISNET: **i)** the ability to create relationships between diseases according to their phenotypic

similarity (**C1**) and **ii)** the ability to increase/improve the phenotypic information of diseases by means of periodic extractions of knowledge (**C2**).

The scenario C1 implies obtaining data for two diseases, which we suspect may share symptoms; we will here use "Influenza" and "Gastroenteritis". The resulting DISNET queries are:

1. disnet.ctb.upm.es/api/disnet/query/**disnetConceptList**?source=**wikipedia**&version=**2018-08-15**&diseaseName=**Influenza**&matchExactName=**true**
2. disnet.ctb.upm.es/api/disnet/query/**disnetConceptList**?source=**pubmed**&version=**2018-04-03**&diseaseName=**Influenza**&matchExactName=**true**
3. disnet.ctb.upm.es/api/disnet/query/**disnetConceptList**?source=**wikipedia**&version=**2018-08-15**&diseaseName=**Gastroenteritis**&matchExactName=**true**
4. disnet.ctb.upm.es/api/disnet/query/**disnetConceptList**?source=**pubmed**&version=**2018-04-03**&diseaseName=**Gastroenteritis**&matchExactName=**true**

We have here used the DISNET query "**disnetConcepList**", which allows retrieving the list of "**DISNET Concepts**" associated with a given disease. The parameters of this query include: "**diseaseName**", with the name of the disease; "**matchExactName**", to indicate that the search by disease name is exact; and "**source**" and "**snapshot**", to respectively indicate the source and snapshot we want to consult. In this case, we selected to consult the two sources Wikipedia and PubMed, and respectively the snapshots of August 15th, 2018 and April 3rd, 2018. Note that the result will consists of four total lists, two for each disease. To illustrate, Fig. 10 shows an extract of the response from the query (3).

As for the scenario C2, it requires retrieving data for a disease whose list of symptoms may have changed with time, i.e., either increased or decreased. As an example, we considered the disease "Acrodynia", and executed the following DISNET queries:

1. disnet.ctb.upm.es/api/disnet/query/**disnetConceptList**?source=**wikipedia**&version=**2018-02-01**&diseaseName=**Acrodynia**&matchExactName=**true**
2. disnet.ctb.upm.es/api/disnet/query/**disnetConceptList**?source=**wikipedia**&version=**2018-02-15**&diseaseName=**Acrodynia**&matchExactName=**true**

Note that, as in C1, we have here used the query "**disnetConceptList**"; nevertheless, we have here executed it twice, on the same disease (**Acrodynia**) and two different snapshots: February 1st, 2018 and February 15th, 2018.

## Visualization of the result of the DISNET queries

Once the results of the query have been retrieved, the next natural step is their visualization; while the actual output format may vary according to the needs of each specific project, for the sake of clarity we here created a graph representation by using the external tool Cytoscape (*Cyt, 2018*). In both scenarios (i.e., C1 and C2) we generated relationships between diseases and their symptoms, with the aim of visualizing the value and scope of the medical data stored and processed by DISNET. In Fig. 11B we see the relationship between the Influenza and Gastroenteritis diseases on one hand (highlighted in white rectangles), and the set of symptoms on the other. Symptoms were obtained from two different sources, specifically Wikipedia and PubMed: relationships are then respectively represented by red and blue edges. Common symptoms are merged by the layout algorithm in the center

```
"diseaseId": "DIS006504",
"name": "Influenza",
"url": "http://en.wikipedia.org/wiki/Influenza",
"disnetConceptsCount": 38,
"disnetConceptList": [
    {
        "cui": "C0009443",
        "name": "Common cold",
        "semanticTypes": [
            "dsyn"
        ]
    },
    {
        "cui": "C0010200",
        "name": "Coughing",
        "semanticTypes": [
            "sosy"
        ]
    },
    {
        "cui": "C0027424",
        "name": "Nasal congestion (finding)",
        "semanticTypes": [
            "sosy"
        ]
    },
    {
        "cui": "C0231221",
        "name": "Asymptomatic",
        "semanticTypes": [
            "fndg"
        ]
    },
    {
        "cui": "C0015967",
        "name": "Fever",
        "semanticTypes": [
            "fndg"
        ]
    },
    {
        "cui": "C0030193",
        "name": "Pain",
        "semanticTypes": [
            "sosy"
        ]
    },
    {
        "cui": "C0085593",
        "name": "Chills",
        "semanticTypes": [
            "sosy"
        ]
    },
    {
        "cui": "C0231218",
        "name": "Malaise",
        "semanticTypes": [
            "sosy"
        ]
    },
```

**Figure 10  Resulting network from DISNET data.** (A) Network of the medical concepts associated with Acrodynia in Wikipedia; (B) network of shared medical concepts between gastroenteritis and influenza.

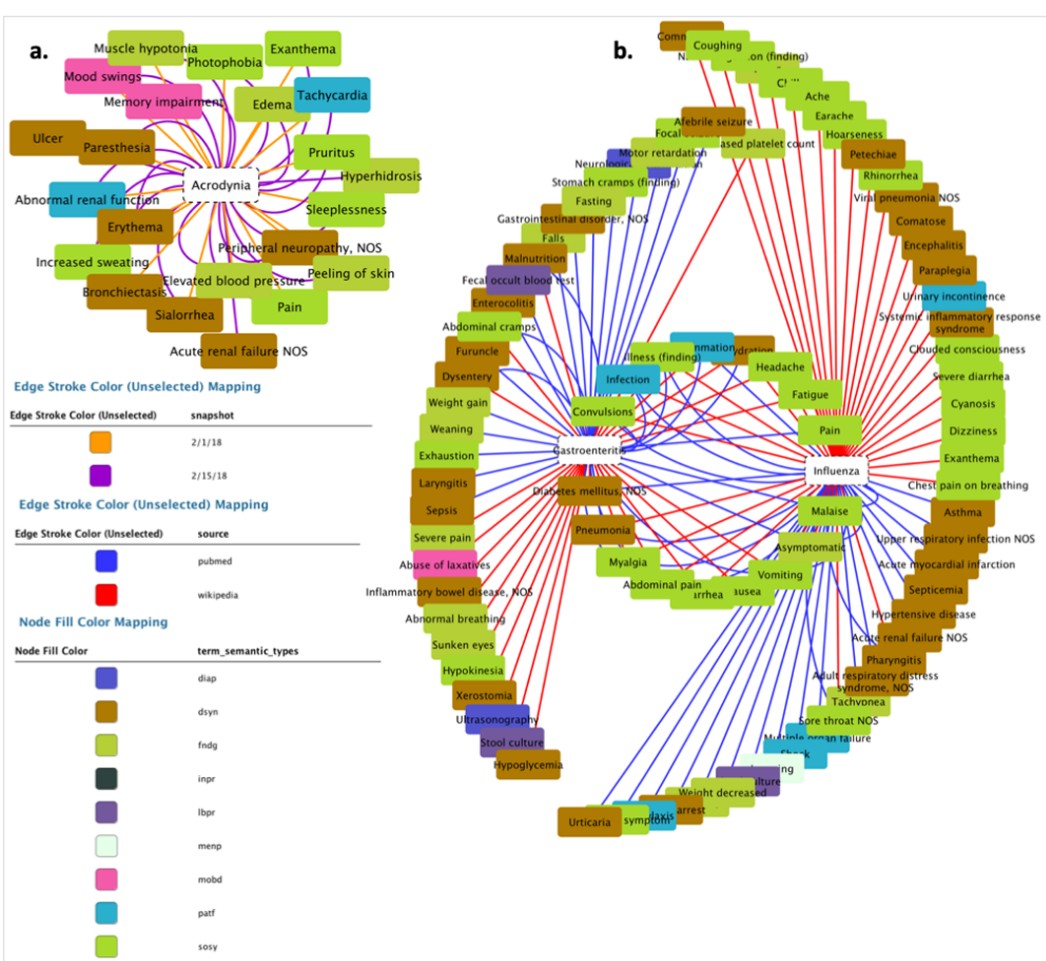

**Figure 11   Response of the system.** The figure is related to the query "disnetConcepList": see example C1.(1) in the main text.

of the graph; the medical terms that are not common among the two diseases, on the contrary, form a peripheral shell. Note that "**Influenza**" has 59 DISNET Concepts and "**Gastroenteritis**" has 47, 19 of which are in common.

In Fig. 11A we observe the network representation of the disease "**Acrodynia**" and of its 18 medical terms, 15 of which were found in the snapshot of February 1st, 2018 and three new ones in that of February 15th, 2018. This is thus an example of an increase in phenotypic knowledge.

This simple use case illustrates how the DISNET system allows generating a network of diseases and their symptoms on a large scale, and that it provides the right environment to know how diseases are related according to their phenotypic manifestations. By applying similarity algorithms, such as Cosine (*Van Driel et al., 2006*; *Li et al., 2014*; *Zhou et al., 2014*) or the Jaccard index (*Hoehndorf, Schofield & Gkoutos, 2015*), it is possible to estimate the similarity between two diseases, and thus to focus further medical analyses on those

pairs showing a large overlap. These features will be also implemented as native features in next DISNET release.

## DISCUSSION

This work presented the DISNET system, starting from its underlying conception, up to its technical structure and data workflow. DISNET allows retrieving knowledge about the signs, symptoms and diagnostic tests associated with a disease. It is not limited to a specific category (all the categories that the selected sources of information offer us) and clinical diagnosis terms. It further allows to track the evolution of those terms through time, being thus an opportunity to analyse and observe the progress of human knowledge on diseases. Finally, it is characterized by a high flexibility, such that new information sources can easily be included (provided they contain the appropriate type of information). We also presented the DISNET REST API, which aims at sharing the retrieved information with the wide scientific community. We further discussed the validation of the system, suggesting that it is good enough to be used to extract diseases and diagnostically-relevant terms. At the same time, the evaluation also revealed that improvements could be introduced to enhance the system's reliability.

## CONCLUSIONS

Among the potential lines of future works, priority will be given to increasing the number of information sources, by including other web sources like Medline Plus or CDC. In parallel, the interested researcher will find in an online repository the instructions to incorporate a new source to DISNET (*DISNET, 2019j*), including the standard structure of the process for incorporating new texts into the DISNET dataset.

Secondly, we are considering the possibility of extending the TVP procedure, by adding new data sources, with the aim of increasing the number of validation terms and hence of reducing the number of false negatives. Note that this could also partly be achieved by resorting to a different NLP tool to process the input texts, as for example to Apache cTakes (*Savova et al., 2010*). Other potential options for future work are the improvement of the ambiguity of medical terms and the implementation of tools that allow the representation of the knowledge extracted and generated. In this context, it is important to note that, currently, the definitions of our medical terms for disease, symptoms and others, are mapped with the vocabularies used by MetaMap. Still this solution has the limitation that these definitions may not be homogeneous with respect to other coding systems or vocabularies. Furthermore, the use of different sources might lead to apparent inconsistencies, like for instance the fact that a same disease could be defined by different sets of symptoms. This problem is intrinsic to the information contained in the sources. Still, DISNET allows to work with the whole information and leaves to the researcher the task of solving such inconsistencies.

Also, future implementations of DISNET also aim to provide ways to automatically compute the similarity between diseases (by using already mentioned and well-known

similarity metrics), extending the DISNET platform to include biological and drug information and developing new visualization strategies, among others.

Finally, increasing the number of queries available from the DISNET API is an essential task to consider for future work, along with the semantization of the complete dataset through the adaptation of the data in DISNET to Resource Description Framework (RDF).

### Funding

The article is a result of the project "DISNET (Creation and analysis of disease networks for drug repurposing from heterogeneous data sources applied to rare diseases)" with grant number "RTI2018-094576-A-I00" from the Spanish Ministerio de Ciencia, Innovación y Universidades. Gerardo Lagunes-Garcia's work is supported by the Mexican Consejo Nacional de Ciencia y Tecnología (CONACYT) (CVU: 340523) under the programme "291114 - BECAS CONACYT AL EXTRANJERO". Lucia Prieto-Santamaría's work is supported by "Programa de fomento de la investigación y la innovación (Doctorados Industriales")) from Comunidad de Madrid (grant IND2019/TIC-17159). The funders had no role in study design, data collection and analysis, decision to publish, or preparation of the manuscript.

### Grant Disclosures

The following grant information was disclosed by the authors:
DISNET: RTI2018-094576-A-I00.
Spanish Ministerio de Ciencia, Innovación y Universidades.
Mexican Consejo Nacional de Ciencia y Tecnología (CONACYT): CVU: 340523.
Programa de fomento de la investigación y la innovación (Doctorados Industriales): IND2019/TIC-17159.

### Competing Interests

Massimiliano Zanin is an Academic Editor for PeerJ.

### Author Contributions

- Gerardo Lagunes-García conceived and designed the experiments, performed the experiments, analyzed the data, prepared figures and/or tables, authored or reviewed drafts of the paper, and approved the final draft.
- Alejandro Rodríguez-González and Massimiliano Zanin conceived and designed the experiments, analyzed the data, prepared figures and/or tables, authored or reviewed drafts of the paper, and approved the final draft.
- Lucía Prieto-Santamaría performed the experiments, authored or reviewed drafts of the paper, and approved the final draft.
- Eduardo P. García del Valle performed the experiments, prepared figures and/or tables, and approved the final draft.

- Ernestina Menasalvas-Ruiz conceived and designed the experiments, authored or reviewed drafts of the paper, and approved the final draft.

## Ethics

The following information was supplied relating to ethical approvals (i.e., approving body and any reference numbers):

Universidad Politécnica de Madrid Ethics Committee approved this study on January 29th, 2018.

## Data Availability

The source code of the entire framework is available at Github: https://github.com/disnet-project/.

Data is available at DISNET (http://disnet.ctb.upm.es/) with free registration.

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
