# Peer review of "DISNET: a framework for extracting phenotypic disease information from public sources"

_PeerJ, doi:10.7717/peerj.8580_

## Round 0.1 · original submission · Minor Revisions

While the reviewers were largely positive, there were a number questions/issues that were raised and that should be addressed in the manuscript revision. In addition to questions within the manuscript, there were some aspects of use with the API that were brought out in the reviews that would be worth considering.

Reviewer 1 ·

Basic reporting

This paper presents a framework named DISNET to extract information from diseases and their signs, symptoms, and diagnostic tests from public data sources. The vast amount of digital data about diseases demands new methods and tools to structure and make sense of them. DISNET is a new approach that tries to address these challenges on textual data, based on a combination of rich data sources, text mining, NLP, and modern Web technologies.

The topic is relevant. The paper is clear, well structured, and easy to understand. The Introduction provides a good description of the current landscape and numerous references to previous works. It appears that the raw data used to build the system is not still available (the authors explain that “access to the data will be enabled once the paper is published”), but users can already perform various queries on the data using the DISNET API.

The figures and tables are relevant according to the paper content, but their legends are not clear enough. They should be self-contained; that is, the content of the table or figure should be understandable without reading the text of the manuscript. Also, the font size used in some figures is too small (e.g., Figure 6, Figure 10). Please, increase the font size to make it more readable.

The paper contains some minor grammar mistakes and typos (I have listed some of them in the “General comments for the author” section), and it would benefit from some closer proofreading.

Experimental design

The authors present a validation of the information extraction process and show how the system can be used to perform queries in two very simple scenarios. In future works, the authors should perform a more robust evaluation of the system, focused on assessing the usefulness and usability of the system in a real setting.

Validity of the findings

The authors adequately present the methods used to build the system and show that (and how) it can be used to retrieve information about diseases via the REST API. My main concern in relation with the validity of the findings is the claim that DISNET “is good enough to be used to extract diseases and diagnostically-relevant terms.” The authors should clarify what the meaning of “good enough” is. How can the authors claim that the system is good at performing a task without conducting a comprehensive evaluation with real users? I strongly encourage the authors to clarify or remove any subjective statements such as that one and limit their claims to the results obtained during their validation process.

Additional comments

1. Integrating information from heterogeneous data sources is a very challenging task. However, the authors state that “thanks to its flexibility, new sources could (and will) be incorporated in the future.” Could you please elaborate on (1) what makes the system flexible; (2) what would be the steps needed to add a new data source to DISNET?
2. How are discrepancies or conflicts between data sources solved? For example, suppose that the symptoms for the disease D1 are S1 and S2 according to one source, but S1 and S3, according to a different source. How would users take advantage of contradictory information in these cases?
3. The authors explain that one of the advantages of their work is that Wikipedia is used as the primary source of knowledge. My concern is that some of the information on Wikipedia may be inaccurate or wrong. Could you please explain in more detail why using Wikipedia over other more formal and curated sources is an advantage and not a disadvantage?
4. Even though I do not consider it essential for publication, the paper would substantially benefit from a structured comparison between DISNET and existing approaches. I suggest that the authors include a table comparing DISNET to other existing systems. This table will help the reader get a better picture of the current state of the art and to understand the advantages of DISNET. The current Introduction section contains an extensive list of references to existing works with no explanation for many of them (e.g., lines 75-85, 95-98). Either using a comparison table or a textual analysis, the authors should summarize existing work and clarify how DISNET is different. For example, the authors mention (see lines 154-157) that having a method for information sharing, such as an API, is not common in existing systems. It would be beneficial to know precisely the systems that have/do not have this capability.
5. A structured comparison between different candidate data sources would enrich the paper as well, and it would help the authors to support their choice. However, again, I don't consider this comparison essential for publication.
6. Abstract - Background: What do the authors mean by “customizable disease networks”?
7. Abstract - Methods: “…extracted from Wikipedia and PubMed websites.” Using the term “websites” imposes a limitation on the scope of the work. Consider removing that term.
8. Queries that return a large number of results (e.g., http://disnet.ctb.upm.es/api/disnet/query/diseaseList) need to support pagination to decrease the response time and unnecessary network traffic.
9. L435: Who has performed the manual analysis?
10. By looking at the DISNET queries presented on the use case, it seems that DISNET is limited to free-text queries. Does the system support ontology-based queries?
11. The DISNET queries presented use the parameter matchExactName. Does the system support any other matching methods?

Grammar and typos (this list may not be complete):
L71: facilitate -> facilitated
L76: system -> systems
L141: recovery -> retrieval
L151: encyclopaedia
L215: “…to include the following three web sites…” -> However, the authors only mention i) Wikipedia, and ii) PubMed. What about the third system?
L244: Virtuous -> Virtuoso
L246: Reference error.
L486: Reference error.
L516: Reference error.

Reviewer 2 ·

Basic reporting

Relative to the basic reporting standards of PeerJ, this manuscript is fully compliant and moreover, high quality in its scholarship. Ample and appropriate references will facilitate contextualization by the intended audience of readers, and facilitate appropriate linking of citations. The English language is excellent, clear, professional, and needs no improvement.

Experimental design

The research presented is within the Aims and Scope of PeerJ, to the best of my understanding, as it concerns biomedical research informatics, and proposes a novel methodology, data workflow and resource. Methods are described in detail and well documented, such as Metamap configuration details at http://disnet.ctb.upm.es/apis/disnet. Source code is apparently not shared, however, the API Is well documented, and includes example queries and other data via the group GitLab repo at https://midas.ctb.upm.es/gitlab/disnet/paperdisnet. Regarding the online resource, an API but without GUI, when I have tried to "Sign Up" the following notice appears: "Registration is temporarily disabled. If you have any question please contact with the administrator" However, I was able to register, obtain a key and use the API, so apparently the notice was spurious.

Validity of the findings

The work described, DISNET, is an ambitious project toward an important goal reflecting high levels of knowledge, skill and effort suited to this task. The title and abstract make clear that the goal is "extracting phenotypic disease information from public sources" which differentiates this project from ontology projects which focus on expert curation of disease knowledge. For the intended audience, the focus on the framework and shared methodology is quite significant, and differentiates this work from contributions comprising databases which may have validity and value worthy of publication, but often lack sustainability and reusability. The authors recognize in the introduction the critical importance of sharability, and go on to further emphasize the relationships among diseases as a critical focus of DISNET, and cite the PubMed-based HSDN as a previous effort along these lines. Five advantages over previous research are claimed, and I agree with all. The strongest points in my view are the use of both structured and unstructured data (and semi-structured, the Wikipedia sections), harmonized systematically from heterogeneous sources, all shared effectively via documented API, in short, adherence to FAIR principles.

My biggest concern about this project and paper relates to the semantic difficulties of the concepts "disease", "phenotype", "symptom", and others. This problem is a chronic condition and limits the rigor of biomedical informatics, and particularly the accuracy of mappings among heterogeneous sources. "Knowledge extraction" depends on strong metadata, ontology and semantics. I expect the authors understand this well, and for example the issue of what is or is not a disease. Is "hypertension" a disease? Or symptom? ICD-9 includes "Unspecified essential hypertension" (code 401.9). Diseases are defined in various ways, for various reasons, by different sources. Likewise for symptoms and phenotypes, etc., what is and is not a phenotypic trait or symptom will depend on definitions which can vary among sources. And, the relationships can vary. A symptom may be a diagnostic criterion, thus its presence is somewhat definitional and tautological (e.g. gastritis ⇔ inflammation). A trait could precede and possibly cause disease onset (e.g. hypertension => heart attack), or be a consequence, possibly a complication or another disease (e.g. diabetes => diabetic neuropathy). I am not suggesting any major revision to this manuscript, however, only that these challenges and limitations and understandings be noted, and thereby inform and guide the readers. I am fully in agreement with the use of UMLS and Metamap by the authors in addressing these challenges. I am unfamiliar with TVP and unable to comment without access to the 2015 book and conference paper cited ("Diagnostic Knowledge Extraction from MedlinePlus: An Application for Infectious Diseases").

Additional comments

Congratulations and thank you for the hard work and considerable expertise in producing this resource. In addition to the main concern described, I did experience some timeout errors when using the API (/diseaseList?source=wikipedia&version=2019-10-15). Perhaps this is due to lack of response pagination, as a diseaseList?source=pubmed returns 10+MB and apparently all 2355 diseases. If the goal is to maximize impact and benefit to the community, you might consider allowing downloads, and/or Docker images. Likewise, open source, while not obligatory, may increase use and impact.

---

## Round 0.2 · accepted · Accept

Thank you for addressing reviewer questions and congratulations again.